# Archaeogenomic evidence reveals prehistoric matrilineal dynasty

Douglas J. Kennett[1], Stephen Plog[2], Richard J. George[1], Brendan J. Culleton[1], Adam S. Watson[3], Pontus Skoglund[4], Nadin Rohland[4], Swapan Mallick[4,5,6], Kristin Stewardson[4,6], Logan Kistler[1,†], Steven A. LeBlanc[7], Peter M. Whiteley[3], David Reich[4,5,6] & George H. Perry[1,8]

For societies with writing systems, hereditary leadership is documented as one of the hallmarks of early political complexity and governance. In contrast, it is unknown whether hereditary succession played a role in the early formation of prehistoric complex societies that lacked writing. Here we use an archaeogenomic approach to identify an elite matriline that persisted between 800 and 1130 CE in Chaco Canyon, the centre of an expansive prehistoric complex society in the Southwestern United States. We show that nine individuals buried in an elite crypt at Pueblo Bonito, the largest structure in the canyon, have identical mitochondrial genomes. Analyses of nuclear genome data from six samples with the highest DNA preservation demonstrate mother–daughter and grandmother–grandson relationships, evidence for a multigenerational matrilineal descent group. Together, these results demonstrate the persistence of an elite matriline in Chaco for ∼330 years.

[1] Department of Anthropology, Pennsylvania State University, University Park, Pennsylvania 16802, USA. [2] Department of Anthropology, University of Virginia, Charlottesville, Virginia 22904, USA. [3] Division of Anthropology, American Museum of Natural History, New York, New York 10024, USA. [4] Department of Genetics, Harvard Medical School, Boston, Massachusetts 02115, USA. [5] Broad Institute of MIT and Harvard, Cambridge, Massachusetts 02142, USA. [6] Howard Hughes Medical Institute, Harvard Medical School, Boston, Massachusetts 02115, USA. [7] Peabody Museum of Archaeology and Ethnology, Harvard University, Cambridge, Massachusetts 02138, USA. [8] Department of Biology, Pennsylvania State University, University Park, Pennsylvania 16802, USA. † Department of Anthropology, National Museum of Natural History, Smithsonian Institution, Washington, DC 20013, USA. Correspondence and requests for materials should be addressed to D.J.K. (email: djk23@psu.edu) or to S.P. (email: plog@virginia.edu) or to G.H.P. (email: ghp3@psu.edu).

Major advances in ancient DNA and genomic sequencing approaches have recently transformed our understanding of modern human and archaic hominin admixture[1,2], dynamic population turnover in the European Upper Paleolithic and Neolithic[3–5] and the peopling of the Americas[6,7]. Here we use these methods, in combination with high-precision accelerator mass spectrometry (AMS) [14]C dating, to document the hereditary origins of a long-lasting elite lineage at Pueblo Bonito in Chaco Canyon, New Mexico (USA), the political and ritual centre of one of North America's earliest complex societies. Hereditary leadership formed the basis of the earliest historically recorded complex societies around the world[8], but the cultural evolutionary foundations of these systems are unknown elsewhere because of the absence of writing in societies like Chaco.

Pueblo Bonito was the largest (∼650 rooms, Fig. 1) of more than a dozen, multistoried masonry buildings, referred to as great houses, in Chaco Canyon. From 800 to 1130 CE, remarkable population expansion occurred in much of the Pueblo World, but Chaco was unusual because of the coalescence of some communities into a concentration of great houses ranging in size between 50 and 650 rooms[9]. By the end of the ninth century CE, there were no fewer than three of these multistoried great houses in the canyon, including Pueblo Bonito. Bonito itself and the human presence in the canyon expanded rapidly in the eleventh and early twelfth centuries CE, including the construction of at least 10 new great houses and scores of outlying settlements that adopted Chacoan architectural style[10], some connected by roads indicating significant investment in political and ritual infrastructure (Fig. 1). This was followed by a rapid decline in construction and the almost complete depopulation of the canyon by 1130 CE[9].

Archaeologists increasingly have recognized the complexity of Chacoan society and acknowledged that some individuals had greater power than others[11]. There is little agreement, however, regarding the exact nature or basis of political hierarchy. Embedded within this larger debate is the question of whether Chacoan society was organized similarly to historic Western Pueblo peoples (for example, the Hopi of northeastern Arizona) wherein social groups with a matrilineal core were ranked and controlled ritual sources of power[12] or, alternatively, were led by non-kinship-based ritual sodalities, comparable to the medicine societies of some Eastern Pueblos[13]. In the latter, the highest-ranking members were chosen based more on ability and achievement rather than descent (whether patrilineal or matrilineal). The debate about the importance of kinship and unilineal descent groups in the evolution of complex society is a long-standing issue in anthropology (see, for example, refs 14,15) and has been central to much discussion of the evolution of Pueblo social organization[16,17]. Absent from this history of debate is the most direct method for studying descent in prehistoric populations, the analysis of ancient DNA data from skeletal material recovered during more than a century of excavation in the canyon.

Our effort to address these questions focusses on the extent of familial relationships among individuals interred in the most elaborate burial crypt in the Pueblo World. Pueblo Bonito Room 33 was part of the initial construction phase of this great house in the ninth century CE[11]. Interment inside Chacoan settlements was unusual as otherwise most individuals were buried outside in mounds of domestic debris[11]. The unusual burial pattern combined with an atypical, hatch-like entry in the eastern wall just below the roof of this small ∼2 m × 2 m room suggests that room 33 was purposely constructed as a crypt for a high-status member of this nascent community (burial 14) and ultimately his lineal descendants.

Results from previous osteological analysis indicate that burial 14 was a male in his 40s who died after a lethal blow to the head[18]. His body was placed in the centre of the burial crypt on a prepared floor of sand and wood ashes and adorned with thousands of turquoise (n > 11,200) and shell (n > 3,300) beads and pendants, originally part of necklaces, anklets and bracelets[19,20]. The abundance of turquoise alone makes this the richest burial known in the North American Southwest. Also placed on the right side of the individual were several other unusual offerings including multiple abalone (Haliotis spp.) shells from the Pacific Coast and a conch shell trumpet. The body was then entombed by a thick (70 cm) layer of clean sand devoid of artifacts except for two ceramic vessels. A second individual (burial 13), also associated with large amounts of turquoise (n > 5,800), was buried on top of this thick layer of sand and covered with a unique plank floor.

Twelve other individuals were then buried in this crypt above the wooden floor. When excavated in 1896, most burials above the floor were disarticulated and skeletal elements from different individuals were sometimes commingled, likely a result of disturbance as the bodies of deceased members of this elite lineage were periodically added to the small, dark room, as well as the methods used to excavate the room in 1896. These 12 interments were directly associated with offerings of ceramic bowls and pitchers and in some instances shell and turquoise. Nearby were caches of flutes in the northeastern corner and carved, wooden ceremonial staffs in the ceiling. Turquoise beads were placed around wooden posts in the room corners, a pattern consistent with ethnographic accounts of Puebloan cosmology[11]. Also noteworthy were the multiple caches of ritually important objects in adjacent rooms: hundreds of wooden ceremonial staffs; scores of cylindrical ceramic vessels; the remains of scarlet macaws (Ara macao); and jewelry[20,21].

Here we use archaeogenomic methods, in combination with high-precision AMS [14]C dating, to demonstrate that all nine of the successfully sampled individuals who were buried in this elaborate crypt have identical mitochondrial genomes. Analyses of nuclear genome data from the six individuals with the highest DNA preservation demonstrate mother–daughter and grandmother–grandson relationships consistent with matrilineal descent in this elite lineage. Together, these results indicate that hereditary inequality and societal complexity emerged in Chaco by the early ninth century CE and that this matrilineal dynasty persisted at the centre of this complex society for ∼330 years until its rapid collapse in the early twelfth century CE.

## Results

**AMS [14]C dating**. We modelled the interment date for burials 14 and 13, and the placement of the split plank floor in room 33, using existing AMS [14]C measurements obtained directly from the skeletal remains of these individuals[11] in a phased Bayesian stratigraphic model (Fig. 2a and Supplementary Fig. 1). Bone collagen was extracted from the crania of nine additional individuals interred above the wooden floor and purified using ultrafiltration. Crude and ultrafiltered gelatin yields were excellent and stable isotope measurements are consistent with a population relying on maize agriculture ($\delta^{13}$C: − 7 to − 9.5‰ VPDB; $\delta^{15}$N: 11.1 to 13.3‰ Atm N$_2$, Supplementary Table 1). Burial 14 dates to the early ninth century (800–850 CE) and similar direct dates on abalone shells (Haliotis fulgens) from the southern California coast that were buried in association with this individual suggest that the burial remained intact and unaltered after this time (Supplementary Table 2). Persistent burial of the remaining individuals in this crypt occurred until ∼1130 CE (Fig. 2a). Chronological simulations indicate interments occurred

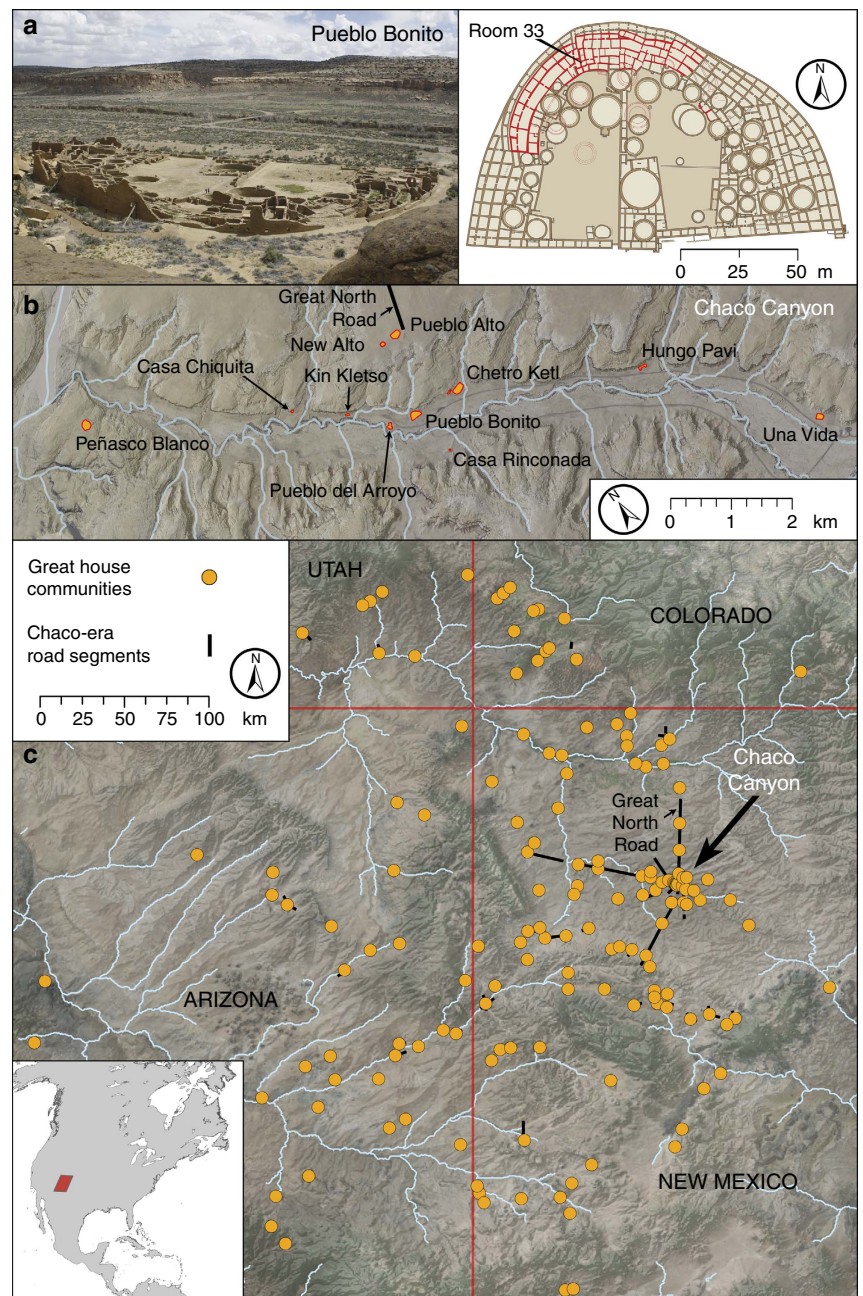

**Figure 1 | Photographs and maps of Pueblo Bonito and surrounding Chacoan sites.** (**a**) Photograph and plan view map of Pueblo Bonito with the location of room 33 noted (photo: G. Perry). (**b**) Locations of the primary great houses in Chaco Canyon. (**c**) Locations of settlements of Chaco Canyon with great house-style architecture with inset showing North America. Maps and graphical layout: T. Harper. Each major element (other than the photograph, labels and legends) was produced in ArcGIS 10.4. All subsequent layout and design were performed in Photoshop CC 14.2.

continuously, rather than episodically, during this ∼330-year period, suggesting persistent transgenerational use of the crypt throughout the growth and decline of Chaco as a regional political and religious centre (Supplementary Figs 2 and 3).

**Mitogenomic sequencing.** We next used archaeogenomic methods to reconstruct the complete or near complete mitochondrial DNA (mtDNA) genomes of nine room 33 individuals for whom ancient DNA preservation levels were sufficient for this analysis, including burials 13 and 14 (Fig. 2a). After sequence read quality and damage filtering (Supplementary Fig. 4), we called per-individual consensus sequences for nucleotide positions covered by a minimum of two independent reads. The resulting consensus sequences averaged $16,231 \pm 455$ bp, or 98.1% of the complete mitochondrial genome, with the average number of reads per nucleotide position (fold-sequence coverage) ranging from $7.4 \times$ to $117 \times$ among the 9 individuals (Supplementary Data 1). All nine room 33 mtDNA genome sequences were identical (Fig. 3 and Supplementary Fig. 5) and belonged to a B2y1 haplotype. The B2 haplogroup has been observed at low frequency among Southwest Native Americans based on hypervariable region sequences[22]. This observation is consistent with the hypothesis that the individuals buried in room 33 were members of a single, elite matriline that played a central leadership role in the Chacoan polity for ∼330 years.

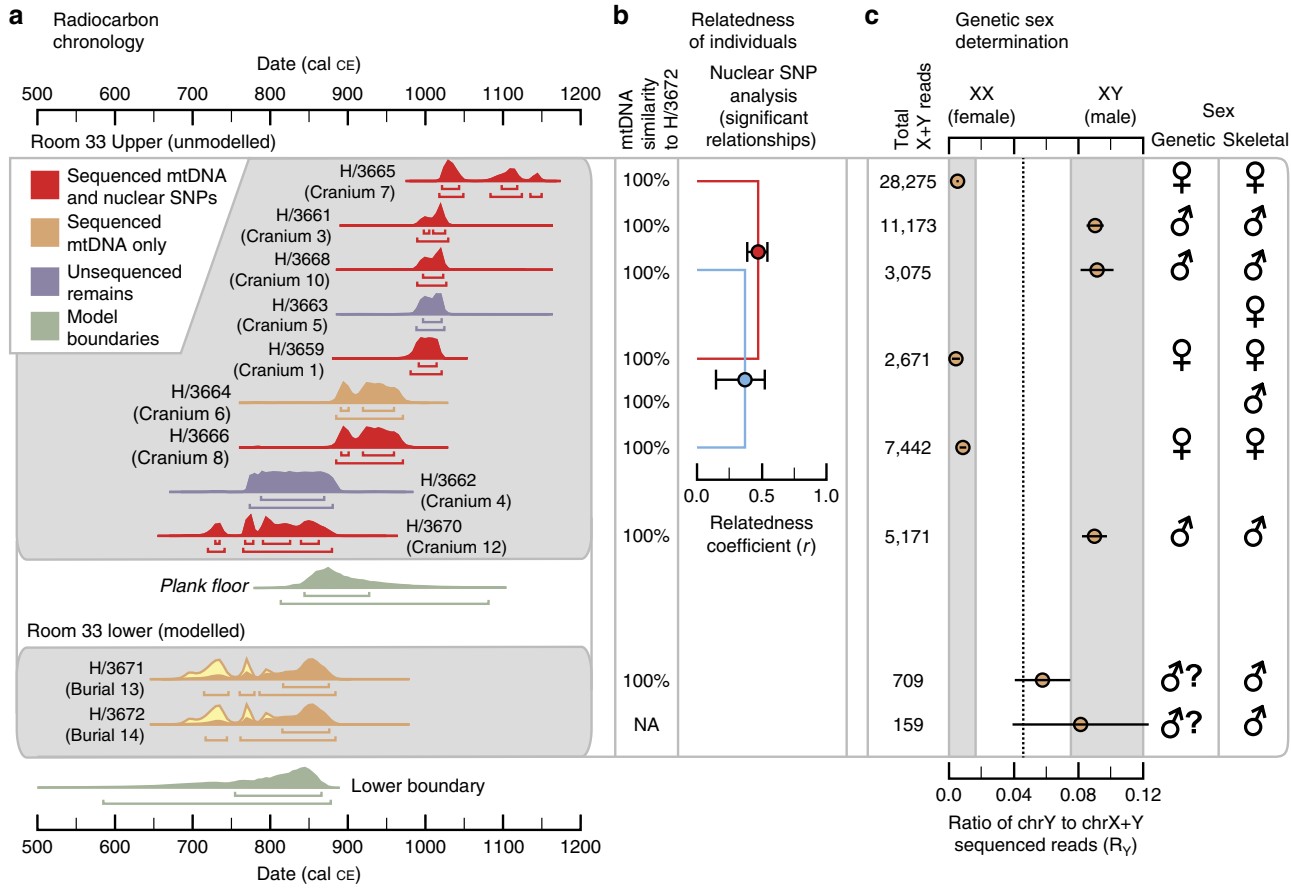

**Figure 2 | Radiocarbon and archaeogenomic results for individuals buried in room 33.** (**a**) Bayesian chronological model of calibrated AMS [14]C dates for burials 13 and 14 and the estimated age of the wood plank floor, and calibrated AMS [14]C dates for nine other crania above the floor in room 33. Calibrated 67.2 and 95.4% AMS date ranges and calibrated posterior probability distributions are illustrated for each cranium. (**b**) Archaeogenomic relatedness results for mtDNA (nucleotide sequence similarity to burial 14) and the nuclear genome. For the nuclear genome analysis, the estimated relatedness coefficient r (with the 95% confidence interval) is shown for each of the two pairs of individuals with significant genetic relationships. (**c**) The ratio Ry of the number of non-redundant sequence reads for each individual that mapped to human chromosome Y to the number of total reads that mapped to chromosomes X and Y, with the total number of X + Y reads and the 95% confidence interval indicated. Shaded areas indicate the Ry boundaries corresponding to confident male and female designations, determined with individuals of known sex[25]. The genetic sex estimates are compared with those from a previous osteological analysis[18]. Graphical layout: T. Harper. The radiocarbon plot on the left was produced in OxCal 4.2 and graphs **b** and **c** were produced in Grapher 10.5. All subsequent layout and design were performed in Illustrator CC 17.1.

**Nuclear genome SNP genotyping and genetic relatedness.** For six room 33 individuals, levels of ancient DNA (aDNA) preservation were sufficient to attempt genotyping a set of >1 million single-nucleotide polymorphisms (SNPs) from across the human nuclear genome[5,23]. We successfully recovered sequence data for an average of $73,247 \pm 46,843$ SNPs per individual (range: 14,948–144,227; Supplementary Data 1) and used the intersecting SNPs for each pair of individuals to estimate the relatedness coefficient r (Supplementary Fig. 6 and Supplementary Table 3). We identified significant genetic relationships between two different pairs of individuals: individuals whose remains have been identified as crania 1 and 7 were likely first-degree relatives, and those identified as crania 8 and 10 were either first- or second-degree relatives (all 'lineal,' as opposed to 'collateral,' relations in the standard terminology of kinship-system analyses; Fig. 2b).

**Genetic sex estimation and aDNA authenticity.** Multiple lines of evidence support the authenticity of our room 33 aDNA results. First, among the sequenced reads we observe the expected patterns of aDNA damage that result from cytosine deamination at fragment ends[24] (Supplementary Fig. 4). Second, for three individuals we generated mtDNA sequencing libraries separately in the ancient DNA laboratories at both Penn State University and the Harvard Medical School. The reconstructed mtDNA sequences were identical between the two labs. Third, for each individual with sufficient data we aligned the sequenced reads from our mtDNA analysis that did not align to the human mtDNA genome to the human reference nuclear genome in order to determine genetic sex from the ratio of chrY to chrX + Y mapped reads[25]. Based on this analysis, three of the sequenced room 33 individuals were classified as female, three as male, and the sexes of the two oldest individuals in the sequence (burials 13 and 14) could not be determined confidently, but were likely male (Fig. 2c). These genetic sex estimates are 100% concordant with the most recent osteological sex determinations of the room 33 burials[18] (Fig. 2c and Supplementary Data 1). These genetic sex estimates were additionally confirmed for six individuals (crania 1, 3, 5, 7, 10 and 12) using the nuclear genome SNP genotype data described above (Supplementary Data 1).

**Discussion**

The use of room 33—constructed as an elaborate burial crypt in what became the largest great house in the region—provides

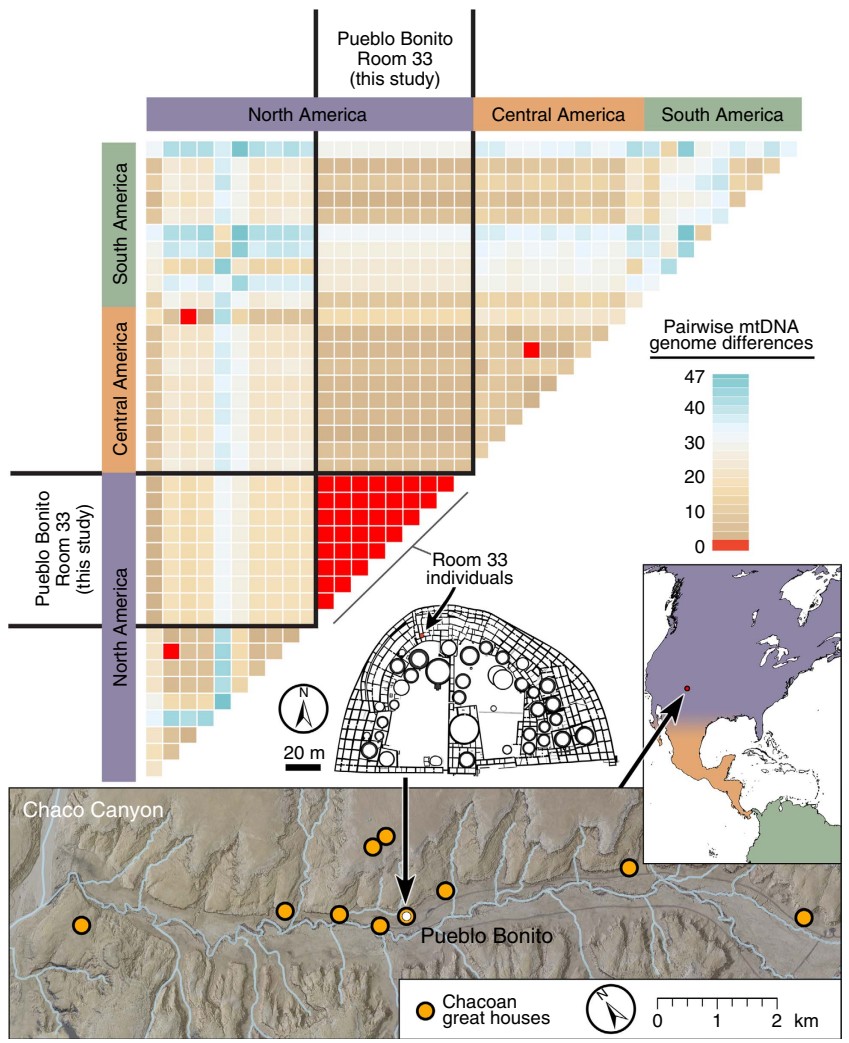

**Figure 3 | Heatmap distance matrix of pairwise mitogenome comparisons.** The number of nucleotide sequence differences between all pairs of 39 Native American complete mtDNA genomes, representing the 9 prehistoric Pueblo Bonito room 33 mtDNA genomes and those previously published for 10 individuals each from North, Central and South America. Positions with missing data or deletions in any individuals were removed from all pairwise comparisons (bp analysed = 14,247). The non-Pueblo Bonito samples from each region were selected at random from an overall database of 171 complete mtDNA genomes (see Supplementary Fig. 5 for a pairwise distance matrix including all samples). (Graphical layout: T. Harper. All cartographic elements were produced in ArcGIS 10.4, with layout and design performed in Photoshop CC 14.2.)

evidence for the early hereditary basis of leadership in the Chaco World. Based on the combination of the relatedness estimates from the nuclear genome data, our high-precision AMS $^{14}$C dates, and osteological and genetic-based estimates of sex and ages of the room 33 individuals, we infer that two generations separated the individuals whose remains have been identified as cranium 8 (30–40-year-old female) and cranium 10 (30–35-year-old male), suggesting a grandmother–grandson relationship. Only one generation separated the individuals identified as cranium 1 (35–45-year-old female) and cranium 7 (23–27-year-old female), suggesting a mother–daughter relationship. Both of these observations are consistent with the importance of close matrilineal ties among those interred in the room (Fig. 4). Although hereditary relationships have been explored elsewhere in historic contexts using short-tandem repeat genotyping[26–28], to our knowledge this is the first study using genome-wide data to document hereditary relationships among individuals within an elite lineage using archaeogenomics, in the absence of a written record, anywhere in the world.

The extravagant grave goods and ritual objects associated with the lineage founder (burial 14) at the base of the room 33 crypt are clear status markers consistent with the elaboration evident in other early complex societies[11,15,29], an indication that a high degree of social differentiation and societal complexity existed in Chaco by the early ninth century. Our work demonstrates that institutionalized hereditary leadership was then passed through the female line until the early twelfth century CE. Matrilineal systems are present ethnographically in all the Western Pueblos (Hopi, Zuni, Acoma, Laguna), and the Rio Grande Pueblos (Cochiti, San Felipe, Santo Domingo and Zia) that speak the Keresan language (see Supplementary Note 1). Although the relationship between prehistoric people from Chaco and specific modern Native American groups remains uncertain, our finding is consistent with a widespread pattern reported for Pueblo social organization beginning with Spanish colonial records[12], and with Pueblo oral traditions recorded since the nineteenth century, especially among the Hopi and Zuni, that emphasize matrilineal clans as the founding social units and matrilineal leaders as key protagonists in social history[12].

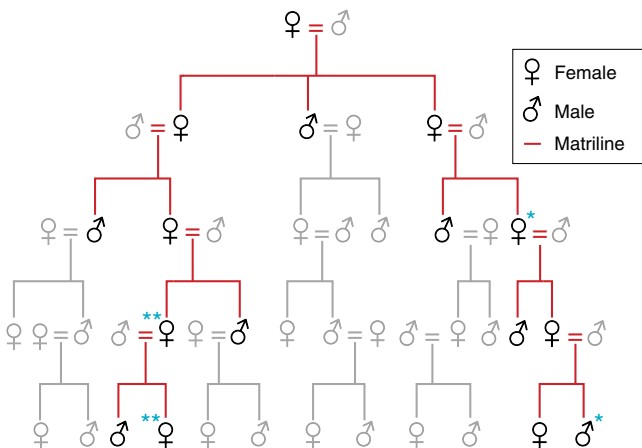

**Figure 4 | Diagram of a hypothetical Chacoan matriline.** The diagram illustrates the inferred relationships for at least two pairs of individuals interred in room 33. Matriline members and descent are highlighted in red. Lineal relationships (mother–daughter and mother–son) and sibling relationships are first degree. The blue * shows the grandmother–grandson relationship suggested for crania 8 and 10 and the blue ** indicates the mother–daughter relationship suggested between crania 1 and 7. Ethnographically, matrilineal systems are present in all the Western Pueblos (Hopi, Zuni, Acoma, Laguna), and also among Rio Grande Keresans (see Supplementary Fig. 7). Graphical layout: T. Harper.

## Methods

We submitted a research proposal to the American Museum of Natural History (AMNH) requesting samples from Chaco Canyon burials classified as culturally unidentifiable following NAGPRA (Native American Graves Protection and Repatriation Act) criteria. The AMNH review committee approved our proposal in accordance with all NAGPRA legal guidelines governing research.

**AMS [14]C and stable isotope analyses.** Methods used for AMS [14]C dating crania 13 and 14 are published elsewhere[11]. In the present study, we directly dated nine crania recovered from above the split plank floor in room 33. Previously published radiocarbon dates for interments above the floor came from long bones[15]. This skeletal material was commingled, and hence we directly AMS [14]C dated each of the crania to verify the date of interment.

Dentine was extracted from tooth roots for [14]C and stable isotope analyses and was extracted and purified using the modified Longin method with ultrafiltration[30]. Tooth samples were initially cleaned of adhering sediment and the exposed surfaces were removed with an X-acto blade. Samples (200–400 mg) were demineralized for 24–36 h in 0.5 N HCl at 5 °C followed by a brief (< 1 h) alkali bath in 0.1 N NaOH at room temperature to remove humates. The residue was rinsed to neutrality in multiple changes of Nanopure $H_2O$, and then gelatinized for 12 h at 60 °C in 0.01 N HCl. Resulting gelatin was lyophilized and weighed to determine percent yield as a first evaluation of the degree of bone collagen preservation. Rehydrated gelatin solution was pipetted into precleaned Centriprep 30 ultrafilters (retaining > 30 kDa molecular weight gelatin) and centrifuged 3 times for 20 min, diluted with Nanopure $H_2O$ and centrifuged 3 more times for 20 min to desalt the solution. Carbon and nitrogen concentrations and stable isotope ratios were measured at the Yale Analytical and Stable Isotope Center with a Costech elemental analyzer (ECS 4010) and Thermo DeltaPlus analyzer. Sample quality was evaluated by % crude gelatin yield, %C, %N and C/N ratios before AMS [14]C dating. C/N ratios for all samples fell between 3.11 and 3.16, indicating good collagen preservation[31]. Shell samples were etched in dilute HCL and then hydrolysed in vacutainers with 85% phosphoric acid for 1 h to generate $CO_2$.

Collagen samples (~ 2.1 mg) were combusted for 3 h at 900 °C in vacuum-sealed quartz tubes with CuO and Ag wires. Sample $CO_2$ from collagen and marine shell was sent to KCCAMS where it was reduced to graphite at 550 °C using $H_2$ and a Fe catalyst, with reaction water drawn off with $Mg(ClO_4)_2$ (ref. 32). Graphite samples were pressed into targets in Al cathodes and loaded on the target wheel for AMS analysis. The [14]C ages were corrected for mass-dependent fractionation with measured $\delta^{13}C$ values,[33] and compared with samples of Pleistocene whale bone or calcite (backgrounds, > 48 [14]C kyr BP), late Holocene bison bone (~ 1,850 [14]C BP), late AD 1800s cow bone and OX-2 oxalic acid standards for calibration. Results are presented in Supplementary Table 1.

**Calibration and modelling.** AMS [14]C dates below the split plank floor were calibrated with OxCal version 4.2.3 (ref. 34) within a stratigraphic model using the IntCal13 northern hemisphere curve[35]. Using *Sequence* and *Phase* models in

OxCal, a rough ordering of depositional events in room 33 below the split plank floor was used to help constrain the calibrated ages of for crania 14 and 13. The initial construction or preparation of the room for use as a crypt is modelled with a *Boundary* followed by a *Phase* ('room 33 lower') including crania 13 and 14, below the plank floor. Two directly AMS [14]C dated abalone (*Haliotis* spp.) shells were also included in this model. The emplacement of the plank floor is modelled as a *Boundary* followed by a *Phase* ('room 33 upper'). Crania about the floor were not modelled because of disturbances during subsequent interment events and they cannot be confidently placed in stratigraphic order with respect to each other. Calibrated results are presented in Supplementary Table 2 and Supplementary Fig. 1. The OxCal code used for the overall model is provided in Supplementary Note 2.

The abalone shells were modelled using the Marine13 curve[35] and a reservoir (ΔR) correction of 234 ± 23 [14]C year assuming collection somewhere on the southern California coast in the vicinity of San Diego where this species is concentrated[36].

**Radiocarbon simulations.** The apparent episodic structure in the calibrated ages for the room 33 crania ($n = 11$) could indicate several periods of interment or potentially continuous interment over the course of three centuries. When plotted against the IntCal13 curve[35] (Supplementary Fig. 2), many of the dates fall on reversals and plateau in the calibration curve between 900 to 1150 CE, similar to observed patterns for directly dated macaw remains from Pueblo Bonito as reported elsewhere[21]. Conventional ages that occur within plateaus in the calibration curve cause the artificial stacking or overrepresentation of certain calibrated dates within certain intervals, whereas conventional dates falling on steeper parts of the curve tend to be underrepresented.

The potential biases introduced by the calibration curve were explored by simulating radiocarbon ages from 750 to 1250 CE in OxCal v4.2.3 (refs 34,37). The *R_Simulate* command specifies a calibrated age and a measurement error, translates the age through the curve to generate a conventional age BP and then calibrates the conventional age. A model was made to simulate 10 dates with a precision of ± 20 [14]C years every 25 calendar years from 750 to 1250 CE for a total of 210 simulated dates. The conventional ages were extracted and frequencies plotted in 20 [14]C-year increments (Supplementary Fig. 3).

Comparing the frequency of simulated [14]C ages with the expected (40 per century), it would be anticipated that even with more or less continuous (for example, generational) interment of individuals in room 33 that fluctuations in the calibration curve produce an apparent episodic structure on the conventional and calibrated ages through the modelled period. Given that the sample size ($n = 11$) is fairly small compared with the overall period of interment (~ 330 cal years), it is difficult to argue that certain periods are over- or underrepresented, as was possible with the procurement of macaws over a shorter interval[21]. For room 33, we do observe that a few gaps in the [14]C ages do correspond to periods where the calibration curve is relatively steep, and therefore are likely artefacts of the curve rather than pauses in interment. Overall, the simulations indicate the observed [14]C ages of room 33 are consistent with continuous rather than episodic use of the crypt from the ninth through the twelfth century CE.

**Ancient mtDNA capture and sequencing.** Skeletal remains from nine individuals from room 33 (Supplementary Data 1) were successfully analysed at the ancient DNA facilities at Pennsylvania State University and Harvard Medical School. Eight of the nine room 33 samples for which archaeogenomic data are analysed in this study (excepting cranium 6) were processed initially in the Penn State ancient DNA facility, with the mtDNA genome capture and sequencing performed as described in this section. The mtDNA genome sequences for three of these same individuals (crania 1, 3 and 7) were also determined at the Harvard Medical School ancient DNA facility, as part of the screening process for the nuclear genome SNP genotyping analyses that were performed (see below). The mtDNA genome of one room 33 individual, cranium 6, was sequenced only at Harvard Medical School.

At Penn State, DNA extractions and pre-PCR library preparation steps were performed in a dedicated sterile laboratory with high-efficiency particulate arrestance-filtered air and positive air flow. Strict procedures were followed to prevent contamination and provide a sterile laboratory environment. All surfaces and workstations were sterilized using a concentrated bleach solution and 70% ethanol solution, before and after each use. Equipment and consumables were irradiated under ultraviolet light for 60 min before samples were introduced. Dedicated reagents were only opened in a sterilized ultraviolet hood located within the clean laboratory. Negative control reactions were included alongside all DNA extractions and library preparations. DNA extractions and negative control reactions were prepared and processed in tandem.

Individual ancient samples were initially pretreated in 1% bleach solution in a separate nonmolecular laboratory before entering the dedicated sterile laboratory. To further reduce surface contamination, the surface layer of each sample was removed using a Dremel tool fitted with a disposable rotating disc that had been treated in 1% bleach solution, followed by multiple washes in molecular grade water to remove traces of bleach. After surface treatment, the samples were ground into a fine powder using a mikro-dismembrator (Sartorius) ball mill with tungsten carbide balls. For each sample, we extracted DNA from 150 to 300 mg of bone

powder exactly following a silica absorption/spin column method[38] and eluted DNA twice in 50 μl TE buffer with 0.05% Tween-20.

Double-stranded libraries with barcoded adapters were constructed with 50 μl of each ancient DNA extract based on the approach by Meyer and Kircher[39]. Before the amplification step of this protocol, the libraries were split into two 10 μl aliquots, with different barcode indices added to each aliquot (Supplementary Data 1). Reactions were prepared in the ancient DNA facility, sealed and then amplified in a separate lab. All libraries were amplified in a 50 μl reaction consisting of PCR buffer, 2 mM MgSO₄, 200 μM each dNTP, 200 nM primer IS4 (5′-AATGA TACGGCGACCACCGAGATCTACACTCTTTCCCTACACGACGCTCTT-3′), 200 nM barcoded p7 primer (5′-CAAGCAGAAGACGGCATACGAGATxxxxxx GTGACTGGAGTTCAGACGTGT-3′, where 'xxxxxx' represents the sample-specific barcode) and 2.5 U Platinum Taq High Fidelity DNA Polymerase (Thermo Scientific). Libraries were amplified as follows: 5 min at 94 °C; 24 cycles of 20 s at 94 °C, 15 s at 60 °C and 20 s at 68 °C; final extension 5 min at 60 °C. Indexed PCR products were purified using SPRI beads and eluted in 15 μl TET buffer.

DNA libraries were enriched for mtDNA fragments using an in-solution biotinylated RNA bait hybridization method[40]. We designed a probe set of 100-mer baits with 10 bp tiling (that is, a new bait starting every 10 bp) complementary to the sequences of five human mitochondrial reference sequences representing the different mtDNA haplogroups observed among Native Americans (GenBank accessions EU095194.1 (A2), EU095219.1 (B2), EU095222.1 (C1), EU095232.1 (D1) and EU095242.1 (haplotype X2a)). The baits were synthesized by MyCroarray, Inc. (probe design: 140429).

The two separately indexed libraries from each sample were combined before hybridization capture. DNA captures were performed using the MYcroarray MYbaits system based on the manufacturer's protocol (Version 2.3) with two modifications: (1) up to 1 μg of the amplified DNA library was used for each sample, and (2) twice as many wash steps were completed following hybridization to remove more nontargeted DNA fragments. Post-capture libraries were amplified using IS5 and IS6 primers[39] and Kapa Hifi DNA Polymerase, otherwise following the MYbaits protocol. After amplification, libraries were purified with SPRI beads and eluted in 15 μl TET buffer.

Post-capture barcoded libraries were pooled and sequenced at the Penn State Huck Institutes Genomics Core Facility on an Illumina HiSeq 2500 platform using 76-bp paired-end reads in Rapid Run mode. Forward and reverse sequence reads were trimmed of adapter sequences and overlapping reads were merged[41], enforcing a minimum 11-nt overlap and base quality score of 20. Merged sequences were mapped to a human reference mitogenome (GenBank accession EU256375.1) using the BWA-backtrack algorithm in the Burrows–Wheeler Aligner, version 0.7.5 (ref. 42). Some samples were sequenced on multiple lanes to increase sequence coverage. All BAM files from these samples were merged to a single per-sample alignment using the SAMtools merge command. Potential PCR-duplicated reads were then removed using the rmdup command from SAMtools-0.1.19 (ref. 43. Reads <20 bp were removed to limit error from nonspecific mapping of exogenous DNA fragments.

**Ancient DNA damage analysis.** Characteristic damage patterns in DNA degradation over time can be used to assess the authenticity of ancient DNA samples[24,44]. Ancient DNA damage is detected by observing increases in specific nucleotide misincorporation patterns diagnostic of mismatches in the single-stranded overhanging ends of sequenced DNA fragments. We analysed each alignment using mapDamage 2.0 (ref. 45) (Supplementary Fig. 4), and extracted mean lambda values from the Bayesian estimation of damage characteristics, describing the per-base probability of terminating an overhang creating a geometric distribution of overhang lengths. We used the cumulative geometric distribution within the R statistical environment[46] to calculate the overhang length encompassing 95% of inferred overhangs per value of lambda (6–14 nt; Supplementary Data 1). We hard-masked all 5′ T and 3′ A residues within this interval. We called a consensus sequence using the SAMtools mpileup command and a perl script, enforcing 2× nonredundant coverage and 80% site identity. Mitochondrial genome haplogroups (Supplementary Data 1) were identified using the program Mitomaster[47] that uses the Mitomap information system and Haplogrep[48]. Individuals with the mtDNA B haplogroup have a characteristic 9-bp deletion compared with the reference genome sequence at positions 8,271–8,279 (9-bp sequence CCCCCTCTA)[49–52]. In this region, the consensus sequences for haplogroup B individuals were erroneous because of misalignment and were manually corrected based on visualization of the sequence reads.

To verify support for the consensus haplotype among only reads with deaminated cytosine residues, we used PMDtools[53] to generate BAM files from fragments having a minimum post-mortem degradation (PMD) score of 3, retaining ~15–43% of reads per alignment. We re-ran mapDamage to generate new lambda values, masked the PMD BAM files and called consensus sequences as above. We called consensus sequences as above with 80% site identity, but with an increased 4× minimum nonredundant coverage to account for the preponderance of damaged sites in the PMD alignments. PMD consensus sequences matched the originals at 100% of represented sites.

**Comparison with modern mtDNA genomes from the Americas.** We curated a reference database of 171 modern complete mitochondrial DNA sequences from

the Americas[54–63], restricted to genomes with ethnicity or region of origin information (Supplementary Data 2). The ancient mtDNA genome consensus sequences from the Chaco region generated in this study were aligned to the modern reference sequences using MAFFT[64] and manually verified using Geneious v8.1.8.

**Genetic sex estimation.** The genetic sex of each ancient individual was estimated using the $R_y$ approach[25] that is based on the ratio of chrY to chrX + Y mapped reads. For this analysis we used two approaches. First, for the results shown in Fig. 2 we used the sequence reads obtained following the mitochondrial DNA capture enrichment that did not map to the human mtDNA genome (that is, 'bleed-through' reads, or those reads that were not removed from the library during the washing steps of the DNA capture protocol (DNA capture is not 100% efficient at reducing library representation). The sequence reads from each sample were aligned to the human reference genome hg18 (NCBI36/hg18 human genome assembly) using the BWA-backtrack algorithm in the Burrows-Wheeler Aligner, version 0.7.5 (ref. 42) and processed as above to remove potential PCR duplicate reads. Sequence alignments were restricted to those with a mapping quality score of at least 30. The genetic sexes of six samples could be estimated confidently, whereas for two other individuals our results were consistent with an assignment of male but the hypothesis that these individuals were female could not be rejected based on the 95% confidence interval (Supplementary Data 1).

Second, we applied the same the same $R_y$ genetic sex estimation approach[25] to data from the nuclear genome SNP genotyping experiments described below for the six individuals with the highest levels of endogenous DNA preservation. This analysis was limited to sequence reads that mapped to the targeted chromosome X and Y SNPs with a mapping quality score of at least 37, where clusters of duplicate reads (as identified by orientation, and start and end position) were represented by the read with the highest sequence quality. The sex estimates from this analysis were 100% concordant with those based on the mtDNA capture bleed-through data (Supplementary Data 1).

We also compared the genetic sex estimates for the eight room 33 individuals with two previous osteological sex determinations[18,65], showing 100% concordance with the more recent assessment[18] and resolving the discrepancy for one individual between the two osteological studies (Supplementary Data 1).

**Nuclear genome SNP genotyping.** In a dedicated ancient DNA clean room at Harvard Medical School, we generated nine ancient DNA libraries, either from powder drilled directly from teeth end extracted using a protocol optimized for the recovery of ultra-short DNA fragments[66] or using extracts made in the ancient DNA clean room at Penn State. We converted the extracts into sequencing libraries, most of which were treated with uracil-DNA-glycosylase[67].

We first used in-solution capture methods to enrich the DNA library for human mitochondrial DNA fragments[5,23] to screen for samples with sufficient quality for nuclear genome SNP genotyping. We sequenced the enriched DNA libraries on an Illumina NextSeq500 instrument using 2 × 75 base pair reads. We used Seqprep[68] to identify paired sequences overlapping by at least 15 base pairs, and then mapped these reads to the mitochondrial DNA genome RSRS[56] using BWA-0.6 (ref. 42).

We took forward to genome-wide analysis samples with appreciable amounts of DNA and no evidence of heterogeneity of mitochondrial sequences based on contamMix-1.0.9 (ref. 69). One accepted strategy for ancient DNA authentication in genome-wide ancient DNA analysis is to require at least 10% damage in the first nucleotide for a non-UDG-treated library and at least 3% in the first nucleotide for a partially UDG-treated library, and all libraries we took forward to genome-wide analysis met this requirement[67].

We enriched the libraries taken forward to genome-wide analysis for 1,237,207 SNP targets (SNP panels 1 and 2 from Fu et al.[23] that include all SNPs on the Affymetrix Human Origins, Illumina 610-Quad and Affymetrix 50k arrays). We sequenced the enriched products on a NextSeq500 instrument using 2 × 75 base pairs reads, and processed the sequences as we did for the mitochondrial DNA analysis except that we mapped instead to the human genome reference sequence hg19. We restricted our analysis to six samples for six distinct libraries passing our quality control. These samples had 14,948–144,227 SNPs covered by at least one sequence read.

**Relatedness coefficient estimates.** We extracted 856,473 SNPs on chromosomes 1–22 that are not in CpG dinucleotide context in order to avoid the potential influence of differences in postmortem damage rates among individuals. For each pair of individuals, we computed the average mismatch rate across all autosomal SNPs covered by at least one sequence read for both of the two individuals being compared (when >1 sequence read was present for one individual at a given site, a random read was sampled for the analysis) and computed s.e. using a weighted block jackknife. There were too few X-chromosome SNPs with sequence data for high-confidence relatedness results.

We observed that the greatest mismatch rates x between pairs were ~21%, but several were significantly lower (Z>3), providing putative evidence of relatedness. To estimate the relatedness coefficient r, we need to account for the fact that we are downsampling the data to a single sequence at each position. Thus, if we were comparing two identical individuals, the mismatch rate would still be only half of

that observed between unrelated individuals. We thus correct our estimate of $r$ using the expected mismatch rate because of random sequence sampling $b$. We choose $b = 0.21/2 = 0.105$ based on the approximate maximum mismatch rates observed. Our estimator is thus

$$r = 1 - ((x - b)/b)$$

We also computed a 95% confidence interval using block jackknife standard errors.

**Data availability.** Sequence reads from the mtDNA capture experiments and all nuclear genome SNP genotyping BAM files have been deposited in the NCBI Sequence Read Archive, Accession no. SRP094965 (SRA BioProject no. PRJNA353635). All BAM and FASTA files of both the original and damaged read version consensus mtDNA sequence for each individual have been deposited in the Dryad Digital repository at http://dx.doi.org/10.5061/dryad.3344d. The multisequence alignment has been deposited in the Dryad Digital repository at http://dx.doi.org/10.5061/dryad.3344d.

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

## Acknowledgements

We thank J. Marcus, K. Flannery, D.H. Thomas, C. Stanish, J. Kennett, A. Moore, W. Keegan, R. Bliege Bird, B. Ensor, D. Bird and J. Kantner for useful discussions and comments of previous drafts of this manuscript. D.R. is an Investigator of the Howard Hughes Medical Institute. This project was supported through grants from the National Science Foundation (Archaeometry Program, BCS-1460367, to D.J.K. and B.J.C.), University of Virginia (to S.P.) and Pennsylvania State University (to D.J.K. and G.H.P.). Special thanks to T. Harper for graphical assistance.

## Author contributions

D.J.K., S.P., D.R. and G.H.P. designed research; S.P. performed archival research; R.J.G., B.J.C., N.R., K.S. and L.K. collected data; D.J.K., R.J.G., B.J.C., P.S., S.M., K.S., L.K., D.R. and G.H.P. analysed data, S.P., A.S.W., S.A.L. and P.M.W. provided archaeological and ethnographic interpretations of the results; D.J.K., S.P., R.J.G., B.J.C., D.R. and G.H.P. wrote the paper, with contributions from all authors.

## Additional information

**Competing financial interests:** The authors declare no competing financial interests.

