## [Peer Review File · Nature Communications]

Reviewers' Comments:

Reviewer #1 (Remarks to the Author)

This paper shows evidence from ancient DNA of the burial in the same room of a mother and son and also of a maternal grandmother and grandson, over a period of about 300 years in one of the great houses in Chaco Canyon. It is one of the first cases using just data from ancient dna to do this. The results support evidence from historical records of matrilineal kinship being of great importance in this area. Whilst there are a number of reasons why one might observe an instance of a daughter who might be buried with her mother, or indeed a grandson with his matrilineal grandmother that do not necessarily suggest full matrilineal kinship, the combination of this data with historical records, and in particular other archeological evidence that this family is of especially high status, suggest that this is an important family in a society where less important individuals were not buried in this way, which suggest that a matrilineal group household was of key importance in this society over some period of time.

I am not a geneticists, so cannot really comment on the details of the procedures, except to say that care has been taken to avoid error, replicate some of the analyses in other labs, and bring a range of osteological as well as genetic methods to the table. Hence I am convinced by the conclusions.

The history of kinship is a topic of great interest to anthropologists. This is therefore an important contribution. There are some cases of matrilineal groups having more patrilineality in the very highest status families, so it interesting to see that here this high status family appears to indeed be matrilineal. The paper is clearly written so I have little to add really. Within the limits of my expertise, I do recommend publication.

Ruth Mace

Reviewer #2 (Remarks to the Author)

The authors present a wonderful manuscript, and I applaud that here they present the first study using the new suite of "genomic" methods in ancient DNA research to actually address a question that goes beyond pure large scale population genetics. The described methods are all sound and follow the state of the art in Paleogenomic and Bioarchaeological research. The exciting contexts and relevant anthropological research questions are well described. I think this is all in all a great manuscript and an exciting study and I only have a few minor / but also relevant comments:

Line 102: There must be a typo when it comes to the range of the Delta13C values. In supplementary table 1 the values range from ~7-9.5 not 16 as stated in the manuscript. While ~7 to 9.5 falls well into the range of a diet heavily relying on a C4 diet, 16 would actually be out of that range.

Line 162: I would be careful saying that this is "the first study to document hereditary relationships among individuals within an elite lineage using archaeogenomics". While archaeo"genomics" might already discriminate vs. older studies from the PCR-era for insiders in the field, that might be much less clear for people who are not active in the field of ancient DNA research. There are other examples like Hawas et al 2010, Hummel 2002 where researchers relying mostly on autosomal STRs reconstructed genealogies of "elite" families. The authors here highlight the fact "that they had no "written record" but in case of those other studies written records were also sparse or more confusing than helping. But to my knowledge this is the first study reconstructing kinship relationships using genome wide data from NGS approaches. That would be worth mentioning, because there has been a large gap since the end of the PCR-era with regards to studies addressing kinship relationships, a question archaeologist are often highly interested in, especially in burial contexts with multiple burials. This study shows that the gap is now closed, and that the new archaeogenomic methods can address those relevant questions to the same extend than STR genetic fingerprinting did in the past.

Line 28 abstract: This might be more a question of taste but the last sentence of the abstract leaves a bitter taste of "diffusionism" on my tongue. The highlighted resemblance between the duration of "dynasties" found at Pueblo Bonito and other iconic early complex societies is misleading and unnecessary, and at least methodologically wrong. The results stand on their own without a comparison to a number of other societies that seem randomly picked from all around the world. Since even the discussion does not directly mention this resemblance by name-dropping but takes a far more appropriate and actually excellent way to discuss the findings, I would simply suggest to drop the last 21 words of the last sentence in the abstract.

I recommend this manuscript for publication with minor revisions. At least the two first points discussed should be addressed by the authors, and from an intellectual position the last comment is relevant too, to keep the article in its scientifically relevant context.

Reviewer #3 (Remarks to the Author)

The authors analyse genomic data from 6 individuals as well as complete mtDNAs from three more from a Pueblo Bonito elite crypt from Chaco Canyon, and they found evidence for a matrilineal dynasty persisting during at least 300 years, as well as uncovering kinship relationships between two pairs of individuals.

I think this is a very interesting study that shows, beyond the currently used population genetics approach, the usefulness of archaeogenomic approaches for providing evidences on different scientific aspects such as social and social organisation of past societies.

I found the work interesting, with a cutting-edge methodological approach and a deep archaeological context, very well written and explained to general readers (such as me). This is a great example on how a fruitful collaboration among archaeologists, geneticists and anthropologist can generate new data that would allow an advancement in knowledge on the three fields.

If anything, I found some of the methodological explanations on the ancient DNA labs and anticontamination precautions a bit lengthy and probably unnecessary (eg. positive air pressure, cleaning of bench surfaces with bleach and ethanol, etc). I have been reading the same descriptions for about 20 years now and I think they are already a bit superfluos, especially in the main text. Maybe some of the stuff could be moved to a supplementary description, although of course this is more an editorial decision.

Minor points:

Line 121: "belonged to a B2 haplotype"; In the Suppl Table 1 it seems the haplotype is in fact a B2y (characterised by a C to T mutation at 16261). If this is the case, maybe they could mention the sub-haplogroup itself in the main text, instead of the general "B2"

-Line 146. "...three as male" Does it include the notorious Burial 14? I assume yes, but is not clear in the text, and maybe could be emphasized here.

Genetic sex estimation: I understand from the text that genetic sex was estimated using the left-overs of the mtDNA capture, but on the other hand, the approach followed at Boston for nuclear SNP genotyping (1.2 million) surely include X and Y chromosome SNPs. Where those not used to confirm sex determination?

They use "next generation sequencing (NGS)"; because the third generation technologies are now available, I would rather prefer "second generation sequencing (SGS)", but this is optional of course

Ref 47; I think this reference needs Publisher information

Ref 6: is now published

Point-by-Point Responses to Reviewer Comments

Reviewer #2: states "*The authors present a wonderful manuscript, and I applaud that here they present the first study using the new suite of "genomic" methods in ancient DNA research to actually address a question that goes beyond pure large scale population genetics.*"

Comment #1: Line 102: There must be a typo when it comes to the range of the Delta13C values. In supplementary table 1 the values range from ~-7-9.5 not 16 as stated in the manuscript. While ~-7 to 9.5 falls well into the range of a diet heavily relying on a C4 diet, 16 would actually be out of that range.

Response#1: We have corrected this error. The text now reads:

(Line 117) ($\delta^{13}\text{C}$: -7 to -9.5‰ VPDB; $\delta^{15}\text{N}$: 11.1 to 13.3‰ Atm N₂, Supplementary Table 1)

Comment #2: Line 162: I would be careful saying that this is "the first study to document hereditary relationships among individuals within an elite lineage using archaeogenomics". While archaeo"genomics" might already discriminate vs. older studies from the PCR-era for insiders in the field, that might be much less clear for people who are not active in the field of ancient DNA research. There are other examples like Hawas et al 2010, Hummel 2002 where researchers relying mostly on autosomal STRs reconstructed genealogies of "elite" families.

Response#2: We have clarified this and added two references to previous autosomal STR studies. The text now reads:

(Line 181) Although hereditary relationships have been explored elsewhere in historic contexts using short-tandem repeat genotyping²⁶⁻²⁸, to our knowledge this is the first study using genome wide data to document hereditary relationships among individuals within an elite lineage using archaeogenomics, in the absence of a written record, anywhere in the world.

Comment #3: Line 28 abstract: This might be more a question of taste but the last sentence of the abstract leaves a bitter taste of "diffusionism" on my tongue. The highlighted resemblance between the duration of "dynasties" found at Pueblo Bonito and other iconic early complex societies is misleading and unnecessary, and at least methodologically wrong. The results stand on their own without a comparison to a number of other societies that seem randomly picked from all around the world. Since even the discussion does not directly mention this resemblance by name-dropping but takes a far more appropriate and actually excellent way to discuss the findings, I would simply suggest to drop the last 21 words of the last sentence in the abstract.

Response #3: We agree with this comment and have removed these 21 words from the abstract.

Reviewer #3: states "*this is a very interesting study that shows, beyond the currently used population genetics approach, the usefulness of archaeogenomic approaches for providing evidences on different scientific aspects such as social and social organisation of past societies.*"

Comment #1: If anything, I found some of the methodological explanations on the ancient DNA labs and anticontamination precautions a bit lengthly and probably unnecessary (eg. positive air pressure, cleaning of bench surfaces with bleach and ethanol, etc). I have been reading the same descriptions for about 20 years now and I think they are already a bit superfluous, especially in the main text. Maybe some of the stuff could be moved to a supplementary description, although of course this is more an editorial decision.

Response #1: We have trimmed down the length of the methods, but we feel that this level of detail is needed to replicate the study. This is consistent with the editorial policy of including the methods in the main text.

Comment #2: Line 121: "belonged to a B2 haplotype"; In the Suppl Table 1 it seems the haplotype is in

fact a B2y (characterised by a C to T mutation at 16261). If this is the case, maybe they could mention the sub-haplogroup itself in the main text, instead of the general "B2"

Response #2: This was an error on our part and we have corrected it as follows:

(Line 135) All nine Room 33 mtDNA genome sequences were identical (Fig. 3 and Supplementary Fig. 5) and belonged to a B2y1 haplotype. The B2 haplogroup has been observed at low frequency among Southwest Native Americans based on hypervariable region sequences²².

Comment #2 -Line 146. "...three as male" Does it include the notorious Burial 14? I assume yes, but is not clear in the text, and maybe could be emphasized here.

Response #2: We have clarified this in the text as follows:

(Line 163) Based on this analysis, three of the sequenced Room 33 individuals were classified as female, three as male, and the sexes of two oldest individuals in the sequence (Burials 13 & 14) could not be determined confidently, but were likely male (Fig. 2C).

Comment # 3 Genetic sex estimation: I understand from the text that genetic sex was estimated using the left-overs of the mtDNA capture, but on the other hand, the approach followed at Boston for nuclear SNP genotyping (1.2 million) surely include X and Y chromosome SNPs. Where those not used to confirm sex determination?

Response #3 We have completed this analysis and the data is included in the Supplementary Data file. This analysis is consistent with our other genetic sex determinations. We added the following paragraph to the methods to explain the analysis.

(Line 409) Second, we applied the same the same R_v genetic sex estimation approach²⁵ to data from the nuclear genome SNP genotyping experiments described below for the six individuals with the highest levels of endogenous DNA preservation. This analysis was limited to sequence reads that mapped to the targeted chromosome X and Y SNPs with a mapping quality score of at least 37, where clusters of duplicate reads (as identified by orientation, and start and end position) were represented by the read with the highest sequence quality. The sex estimates from this analysis were 100% concordant with those based on the mtDNA capture bleed-through data (Supplementary Data 1).

Comment #4: They use "next generation sequencing (NGS)"; because the third generation technologies are now available, I would rather prefer "second generation sequencing (SGS)", but this is optional of course.

Response #4: We no longer use "next generation sequencing" in the manuscript, but refer to the specific methods used to help with clarity.

Comment #5 Ref 47; I think this reference needs Publisher information

Response #5: We have added this information

Comment #6: Ref 6: is now published

Response #6: We have updated this reference.

Reviewers' Comments:

Reviewer #3 (Remarks to the Author)

The authors have answered all my points and in my view the paper is acceptable for publication